# Citrus diseases detection using innovative deep learning approach and Hybrid Meta-Heuristic

**Nouman Butt**[1], **Muhammad Munwar Iqbal**[1], **Shabana Ramzan**[2], **Ali Raza**[3], **Laith Abualigah**[4,5,6,7], **Norma Latif Fitriyani**[8], **Yeonghyeon Gu**[8]*, **Muhammad Syafrudin**[8]*

**1** Department of Computer Science, University of Engineering and Technology, Taxila, Pakistan, **2** Department of Computer Science & IT, Government Sadiq College Women University, Bahawalpur, Pakistan, **3** Department of Software Engineering, University Of Lahore, Lahore, Pakistan, **4** Computer Science Department, Al al-Bayt University, Mafraq, Jordan, **5** Centre for Research Impact & Outcome, Chitkara University Institute of Engineering and Technology, Chitkara University, Rajpura, Punjab, India, **6** Applied Science Research Center, Applied Science Private University, Amman, Jordan, **7** Artificial Intelligence and Sensing Technologies (AIST) Research Center, University of Tabuk, Tabuk, Saudi Arabia, **8** Department of Artificial Intelligence and Data Science, Sejong University, Seoul, Republic of Korea

* yhgu@sejong.ac.kr (YG); udin@sejong.ac.kr (MS)

**Data Availability Statement:** The dataset is available at https://www.kaggle.com/datasets/myprojectdictionary/citrus-leaf-disease-image.

## Abstract

Citrus farming is one of the major agricultural sectors of Pakistan and currently represents almost 30% of total fruit production, with its highest concentration in Punjab. Although economically important, citrus crops like sweet orange, grapefruit, lemon, and mandarins face various diseases like canker, scab, and black spot, which lower fruit quality and yield. Traditional manual disease diagnosis is not only slow, less accurate, and expensive but also relies heavily on expert intervention. To address these issues, this research examines the implementation of an automated disease classification system using deep learning and optimal feature selection. The system incorporates data augmentation and transfer learning with pre-trained models such as DenseNet-201 and AlexNet to improve diagnostic accuracy, efficiency, and cost-effectiveness. Experimental results on a citrus leaves dataset show an impressive 99.6% classification accuracy. The proposed framework outperforms existing methods, offering a robust and scalable solution for disease detection in citrus farming, contributing to more sustainable agricultural practices.

## Introduction

Citrus farming is a key player in global agriculture, especially in countries like Pakistan, where it ranks high in fruit production. Citrus fruits, rich in essential nutrients, particularly Vitamin C, contribute significantly to human health. However, citrus crops are prone to diseases such as Anthracnose, Huanglongbing (HLB), Canker, Scabies, Blackspot, and Sandpaper rust, greatly affecting fruit yield and quality. Early detection is vital to prevent crop losses and the spread of infections.

**Funding:** This work was carried out with the support of "Cooperative Research Program for Agriculture Science and Technology Development (Project No. RS-2021-RD010360, Development of pests and plant diseases diagnosis using intelligent image recognition)" Rural Development Administration, Republic of Korea.

**Competing interests:** The authors have declared that no competing interests exist.

The global significance of horticulture is remarkable, as it revolves around the vital aspect of food, which both humans and animals rely on. Therefore, allocating more financial support to the agricultural sector is essential. Citrus fruits are pivotal in agriculture and are widely consumed by almost everyone [1]. The citrus fruit industry has a substantial presence in 137 countries, contributing significantly to the global economy. Citrus fruits are a key component of a healthy diet due to their high vitamin-C content and other beneficial nutrients, offering distinct advantages to human health compared to various other fruits [2]. Several citrus diseases are considered high-risk, including Anthracnose, Huanglongbing (HLB) [3], Canker, Scabies, Blackspot [4], and Sandpaper rust [5]. The manual cycle of citrus sickness discovery is perplexing because of the necessity of devoted time and routine ceaseless checking. Automatic detection of diseases is needed for hours. Image processing techniques are used to address this problem. However, the quality of captured images is inaccurate due to the complex environment and capturing devices.

The application of deep learning in the agriculture sector is an active area of research for addressing disease detection and classification problems. The proposes an automated framework using Deep Learning for citrus disease classification and best feature selection. Citrus diseases can cause significant misfortunes and become liable for financial misfortune because of the diminished creation and nature of citrus organic products. This limitation impacts the useful feature extraction and consequently decreases the classification performance [2]. The usage of these natural products by individuals may cause illnesses. In this way, it is significant for the ranchers to distinguish the illness in the plant before it increases to different parts. Yet, it is a troublesome and tedious interaction to watch out for each plant to identify indications of contamination at the beginning phase [3].

The manual cycle of citrus disease recognizable proof is mind-boggling because of the prerequisite of devoted time and consistent schedule continuous monitoring. In this manner, the off-base illness identifier proportion is high through a manual cycle [4]. The significant complications are vegetation disease recognition and classification. Customarily, vegetation diseases are analyzed in cultivating laboratories [5]. Existing methods and diagnostic tests for citrus infections have fallen short of meeting the agricultural sector's demands. Deep learning can potentially replace manual labor, and electronic tools have been created to both detect and prevent diseases during harvesting. Feature output is a major part of the Computer Vision (CV) [6] area to display the image. The fusion features and algorithm of choice that show a lot of consideration from the previous few years on CV and different techniques develop the visibility of the program [7, 8]. In this paper, we proposed an automatic Hybrid Meta-Heuristic deep learning approach for the classification of citrus diseases.

## Research objectives

The primary contribution is as follows:

- The deep features are generated for each model, which are subsequently refined using the Moth-Flame Optimization Algorithm.

- We enhanced the pre-trained models named DenseNet-201, and AlexNet models were trained using deep transfer learning.

- The proposed model sed the selected features from both deep models are combined using an array-based method and finally classified using supervised learning approaches such as Cost-Sensitive Support Vector Machine (C-SVM), Weighted k-Nearest Neighbors (W-KNN), Quantum Support Vector Machine (Q-SVM), Linear Discriminant Analysis (LDA), Fuzzy k-Nearest Neighbors (F-KNN), k-Nearest Neighbor Bayes (KNB), Multiple

Group Support Vector Machine (MG-SVM), Sparse Discriminant Analysis (SDA), Collaborative k-Nearest Neighbors (Co-KNN), Contextual k-Nearest Neighbors (C-KNN).

- The proposed framework is examined at each phase and compared to recent techniques, affirming its superior performance and compromising results.

The paper is structured in the following manner: The section "Literature Review" provides a review of current and previous techniques in the literature summary of the literature is provided at the end of this section. Section "Proposed Methodology" presents a framework for identifying and categorizing citrus illnesses. It also describes the framework for detecting and classifying diseases of citrus. Section "Results and discussions" contains case studies, results, and discussions to present the outcomes of the case studies and results. Finally, the Section "Conclusion and future direction" covers the conclusion and future suggestions, summarises the findings, and offers further ideas.

## Literature review

In this section, all existing methods of citrus disease recognition are explained. In this work, we used steps including augmentation, modified pre-trained models InceptionV3 [9] and Resnet18, feature extraction [10], GA-based feature selection [11], feature fusion, and final step classification [12]. In recent years, a variety of CV and deep learning methods [13] have been introduced to enhance the detection and identification of plant diseases in the agricultural sector [13]. Over the last decade, agriculture has become a significant focus in CV research [14], with numerous researchers devising techniques for the diagnosis and classification of fruit diseases [15]. These approaches emphasize key stages such as pre-processing [10], partial detection [16], segmentation [17], feature extraction, and ultimate classification.

The author [18] applied a preprocessing technique before any other process to enhance the image and reduce the noise and other factors that directly affect the images, such as lightning, illumination, and occlusion, which may come when capturing images. Various methods of preprocessing have been performed, including image resizing and color transformation, and filters such as median filter, mean filter, etc., for enhancement and noise removal [19]. Utpal et al. [18] proposed two variations of convolutional neural networks (CNNs), namely self-structured (SSCNN) and mobile networks, for separating citrus leaves. In their study, a dataset was first created using Smartphones, and then both models were trained extensively on this dataset consisting of citrus plant images. Mobile Net CNN training has the highest accuracy of 98%, with 92% verification accuracy at 10th. During the 12th, however, the top SSCNN training accuracy was 99% certification accuracy. The suggested method demonstrates that SSCNN is effective in real-time treating citrus leaf disease.

Fangming et al. [20] presented a production approach that utilizes novel Conditional Opposing Auto-Encoders (CAAE) for zero-shot learning, with a focus on Citrus aurantium. The model generates artificial samples from both visual and non-visual classes to improve training balance. The initial citrus disease diagnosis process includes image processing, classification, feature extraction, feature selection, and classification methods. The zero recognition accuracy achieved was 53.4%, which is 50.4% higher than CVAE.

The presented model is used to replace KL-divergence in different auto-encoders and is being explored for a more complex model with additional constraints, such as rotation from visual to semantic space. Sivasubramaniam et al. [21] presented a framework for detecting citrus diseases using deep metric learning, particularly designed for sparse data. This framework processes data using resource-constrained devices like mobile phones. It incorporates a class action-based network patch with distinct modules (focus, collection, and simple neural

network classes) to distinguish various Citrus diseases. Apply a deep metric-based composition by dividing the leaf into parts. The accuracy of both types was 95.04%. They used classification accuracy as a test matrix, and the results were reported after five-fold confirmation. Future work includes the deployment of low-density models to embedded devices. In addition, network parameterization, quantization, and pruning methods can be used to compress deep models.

Farah et al. [22] presented a method called 'small square drop (PLS)' for gathering elements from a set of deeply extracted features. This approach involved combining and selecting algorithms, including convolutional neural networks (CNN), which improved the system's accuracy. The feature extraction method utilized VGG19's two layers (FC6 and FC7) to extract deep features related to depth, color, and texture. The PLS process was also applied in subsequent stages. The integration and selection techniques, when used with PLS, not only increased identification accuracy but also reduced computation time. The method achieved a median accuracy of approximately 90.1%.

Traditionally, disease diagnosis [23] in citrus crops has been slow and prone to errors, requiring expert monitoring, which is labor-intensive and costly. In the last decade, image processing and machine learning (ML), particularly deep learning (DL) [24], have been used to automate disease detection in citrus crops. These methods help reduce labor and improve diagnostic accuracy. However, challenges persist, especially with image quality affected by environmental factors and limitations of conventional imaging devices, which compromise feature extraction accuracy—an essential step in disease detection. Convolutional Neural Networks (CNNs) have shown potential to address these challenges. Models like InceptionV3, ResNet, and MobileNet have been used for plant disease detection [25].

## Research gap

Despite the availability of deep learning methods for citrus disease detection, a gap exists in optimizing feature extraction and selection, critical for improving classification accuracy. Most existing approaches fail to enhance extracted features or include effective feature selection algorithms. Moreover, they often rely on a single model, which may not effectively handle diverse citrus diseases. This paper addresses this gap by proposing a hybrid deep learning framework that combines DenseNet-201 and AlexNet with the Moth-Flame Optimization Algorithm (MFO) for optimal feature selection. By leveraging deep transfer learning and a feature fusion technique, this study aims to enhance classification accuracy and robustness in citrus disease detection.

## Proposed methodology

A potential approach proposed and presents an automated deep framework for identifying Citrus leaf and fruit diseases using deep learning approaches. The structure of this framework is visualized in Fig 1. The procedure initiates with an initial augmentation technique that generates additional training data from the existing samples. Subsequently, two pre-trained models, AlexNet and DenseNet-201, are adapted and fine-tuned through transfer learning using contrast images. Features are extracted from each model, and the feature selection process is carried out using Moth-Flame Optimization. Following feature selection, the chosen features are used for classification through Supervised Learning classifiers such as SVM and K-nearest neighbors (KNN). The framework performance is assessed using challenging datasets and is then compared to existing techniques based on accuracy. Each of these steps is elaborated upon in the following sections.

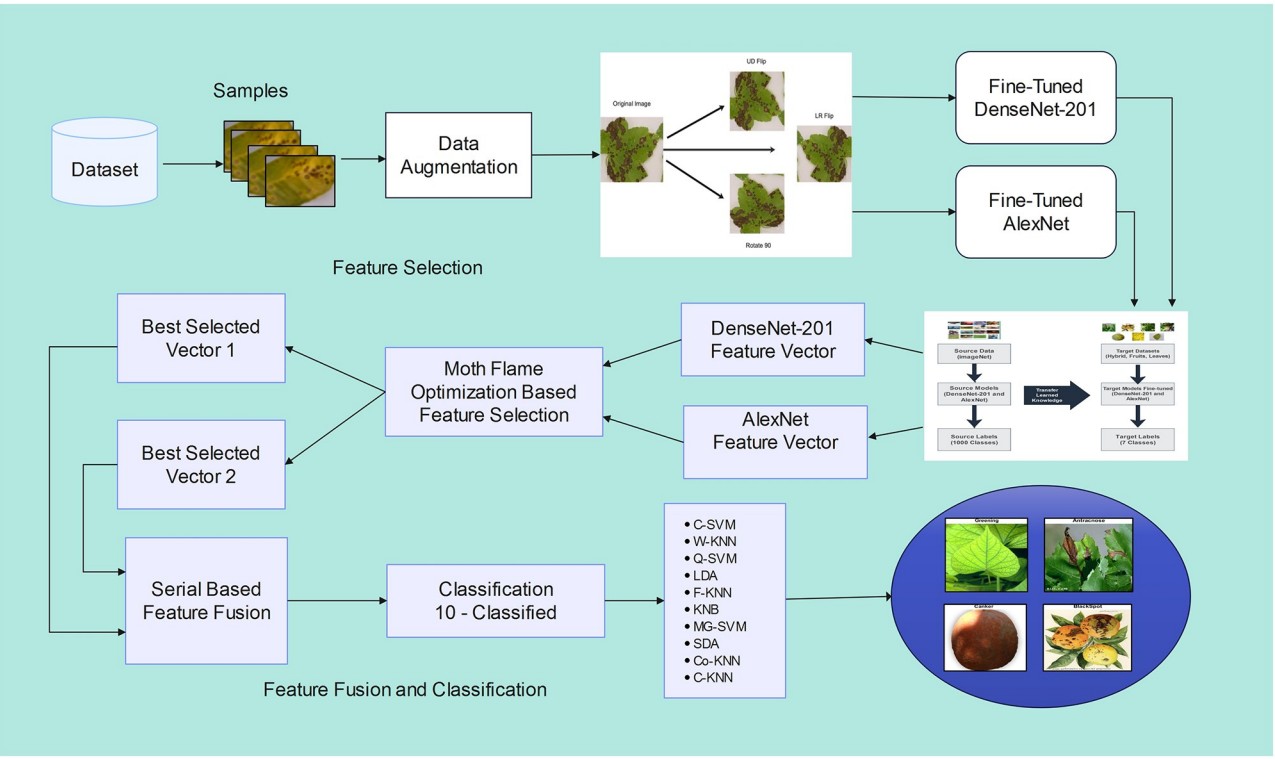

**Fig 1. The proposed framework for citrus disease detection involves a detailed architectural design.**

## Dataset

The experiments in this study were carried out using a publicly available citrus disease dataset. The dataset consists of 7,500 images depicting seven classes of citrus fruit and leaf diseases, including Anthracnose, citrus greening, black spot, canker, melanose, citrus scab, and healthy citrus plants. The dataset was created by combining three different sources: the Hybrid Dataset, the Fruits Dataset, and the Leaves Dataset.

- Hybrid Dataset: Contains 3,988 images, including both fruit and leaf symptoms of various citrus diseases.

- Fruits Dataset: Includes 1,328 images of diseased and healthy citrus fruits.

- Leaves Dataset: Contains 2,184 images of citrus leaves showing different diseases.

## Data augmentation

Data augmentation is a well-known method to reduce over-fitting and enhance model performance. It involves transforming original images while preserving their key features. This technique allows the model to be trained on images from various orientations, improving its performance. It is also useful for addressing class imbalance in classification tasks. Data Augmentation is utilized in this study to mitigate the challenge of having a limited image dataset. More extensive training data typically leads to better deep-learning model performance. Three operations—rotating by 90 degrees, vertical flipping, and horizontal flipping are used. This

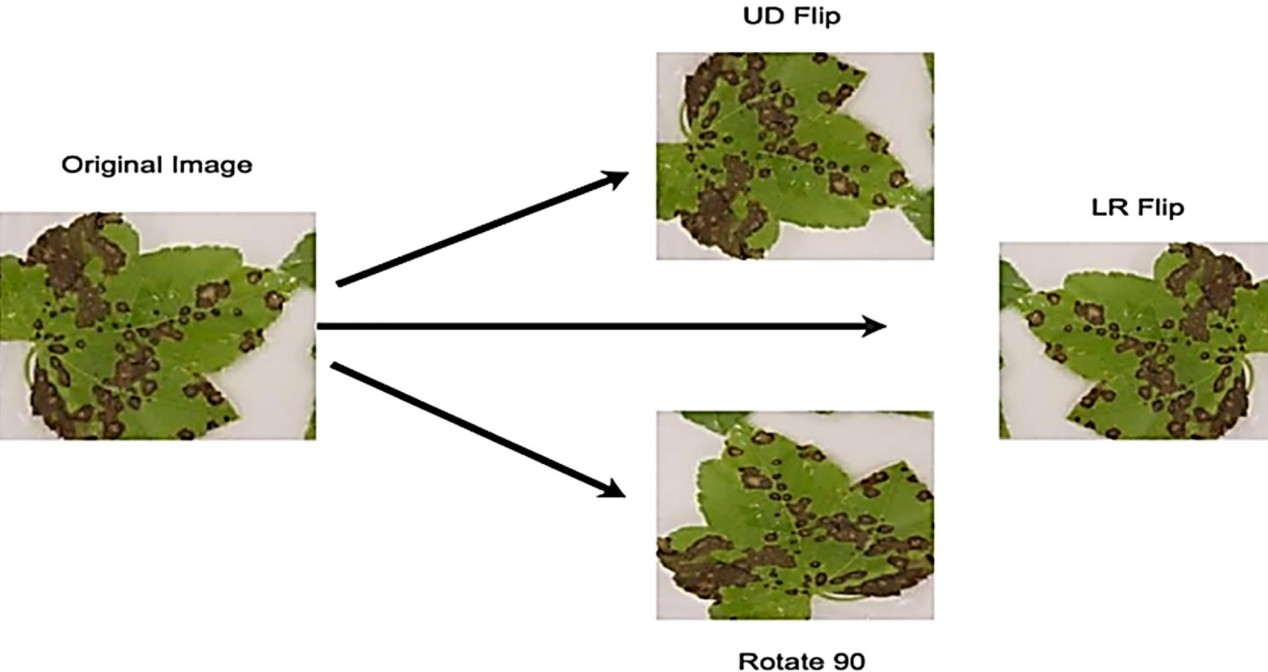

**Fig 2. Data augmentation.** UD: vertical flipping, and LR: horizontal flipping.

study utilizes three datasets: a Hybrid dataset, a Citrus Leaves dataset, and a Citrus Fruit dataset, which initially consisted of 285, 609, and 150 images, respectively. Following the data augmentation procedure, the image counts expand to 3988, 2184, and 1328 for the Hybrid, Citrus Leaves, and Citrus Fruit datasets, respectively. This augmented dataset is then used for model training. The data-augmentation procedure is further displayed in Fig 2.

## Deep feature extraction

**Modified DenseNet201.** DenseNets are gaining popularity and are effective in image processing for object classification. DenseNets, short for densely connected convolutional networks, have similarities to ResNets but also some key differences. Unlike ResNet, which uses an additive approach where the previous output is taken as input for a future layer, DenseNet takes all previous outputs as input for future layers, as shown in the figure. DenseNet was designed to address the issue of vanishing gradients in deep neural networks where information is lost before reaching its destination due to the long distance between input and output layers.

In a DenseNet structure, the output of a layer $a^{[l]}$ is connected to outputs of all previous layers through a function, as expressed in the Eq (1).

$$a^{[l]} = g(a^{[0]}, a^{[1]}, a^{[2]}, \ldots, a^{[l-1]}) \tag{1}$$

Now, let us consider a network with $L$ layers. In a conventional network, $L$ connections correspond to connections between these layers. However, in a DenseNet, there would be approximately $\frac{L(L+1)}{2}$ connections. This means that DenseNets have more connections than other models, even with fewer layers. This unique structure allows DenseNets to train models with over 100 layers efficiently.

**Modified AlexNet.** AlexNet is a groundbreaking convolutional neural network (CNN) mainly used for image recognition and classification. It comprises eight layers with learnable parameters. The model includes five layers with a sequence of max pooling followed by three fully connected layers, all employing ReLu activation, except for the output layer. By utilizing the ReLu activation function, the training speed improved approximately sixfold. Additionally, dropout layers were implemented to mitigate overfitting. The model was trained on the Image-Net dataset, which contains nearly 14 million images distributed across a thousand classes. AlexNet is a deep neural network, and its creators added padding to maintain the size of feature maps during processing. This model's input images are 227x227 pixels and have 3 color channels.

**Feature selection using moth flame optimization.** Nocturnal moths are skilled navigators, using celestial light sources like the Moon and stars for accurate long-distance flight. They typically maintain a straight-line flight path by following a constant angle relative to these celestial bodies, a behavior known as "transverse orientation." However, when moths encounter nearby artificial lights, they become disoriented, mistaking them for celestial objects. Consequently, moths continuously adjust their flight angles in an attempt to maintain a straight trajectory toward the artificial light, resulting in a spiral flight pattern around the light source. In 2015, this unique moth behavior served as the inspiration for the development of the "moth-flame optimization algorithm" by Mirjalili [26]. This algorithm was created to solve complex global optimization problems and was based on a mathematical model that mimicked the moths' spiral flight behavior around artificial lights. In this optimization approach, moth positions change over a series of predefined iterations. During the first iteration, moths are initially scattered randomly within the problem space. The location of every moth is computed using the mathematical expression (2), where $X_{id}$ denotes the $d$th dimension of the position of the $i$th moth, and $Ub_d$ and $Lb_d$ refer to the upper and lower limits for the $d$th dimension, respectively.

$$X_{id} = \text{rand}_{i,d} \times (Ub_d - Lb_d) + Lb_d \quad \text{for} \quad 1 \leq d \leq D \tag{2}$$

In the subsequent iterations, the moths' positions change by being influenced by the position of a "flame." For subsequent iterations, the moths' positions are adjusted according to the location of the flame. The flame index ($R$) is derived using Eq (3), which factors in variables such as the total number of moths $N$ and the maximum iteration count ($MaxIterations$). The flame positions are then established using a step-by-step procedure described in the Table 1.

$$R = \text{rand}\left(\frac{(N - t) \times (N - 1)}{MaxIterations}\right) \tag{3}$$

Finally, when considering the flame number ($R$), each moth can adjust its position using two distinct trials as described in Eq (4). In this equation, $X_i(t + 1)$ represents the new location of the $i$th moth, and $D_i'(t)$ is determined based on Eq (5). The variables $b$ and $k$ are calculated using Eqs (6) and (7), while $F_i(t)$ represents the location of the $i$th flame.

$$X_i(t + 1) = \begin{cases} D_i'(t) \times e^{bk} \times \cos(2\pi k) + F_i(t) & \text{if } i \leq R \\ D_i''(t) \times e^{bk} \times \cos(2\pi k) + F_R(t) & \text{if } i > R \end{cases} \tag{4}$$

$$D_i'(t) = |F_i(t) - X_i(t)| \tag{5}$$

**Table 1. Procedure for constructing flames in the optimization.**

| |
|---|
| **Start with inputs of Moths**: |
| It uses input data including the positions of moths: $X$, |
| Fit: the fitness data values of moths, |
| F: the position of the data flame, and |
| $OF(t)$: the fitness values of flames. |
| **Flame construction in the initial iteration when** $t = 1$. |
| The process involves sorting the Fit vector in ascending order and identifying the sorted indices as $\{j_1, j_2, \ldots, j_N\}$. |
| Construct the flame matrix data $F(t) = \{F_1 \leftarrow X_{j1}, F_2 \leftarrow X_{j2}, \ldots, F_N \leftarrow X_{jN}\}$. |
| **Flame construction for the rest iteration when** $t > 1$. |
| Create a composite matrix *Dual_Pop* by combining matrices $F(t)$ and $X(t - 1)$. |
| Form a vector named vector *Dual_Fit* by combining matrices $OF(t)$ and $Fit(t - 1)$. |
| Sort the vector *Dual_Fit* by ascending order data and extract the sorted index in $\{j_1, j_2, \ldots, j_{2N}\}$. |
| Construct the flame matrix $F(t) = \{F_1 \leftarrow X_{j1}, F_2 \leftarrow X_{j2}, \ldots, F_N \leftarrow X_{jN}\}$. |

$$k = (a - 1) \times \text{rand}(0, 1) + 1 \tag{6}$$

$$a = -1 + t \times \left(\frac{-1}{MaxIterations}\right) \tag{7}$$

In the second trial, which occurs when $i > R$, the parameter $D_i''(t)$ is calculated using Eq (8), and $F_R(t)$ represents the present location of the $R$th flame.

$$D_i''(t) = |F_R(t) - X_i(t)| \tag{8}$$

The Moth-Flame Optimization feature selection algorithm is applied to deep feature vectors obtained from both DenseNet-201 and AlexNet. This process results in the selection of the two best feature vectors, which are combined through array-based concatenation, defined mathematically as follows:

$$Fus(a) = \{Vec(1); Vec(2)\}_{N \times A} \tag{9}$$

In Eq (9), $Fus(a)$ represents the $a^{th}$ feature of a fused feature vector, $Vec(1)$ denotes the first selected feature vector, $Vec(2)$ is the second selected feature vector, and $N \times A$ represents the ultimate size of a concatenated feature vector. This final feature vector is then used as input for supervised learning algorithms in the final classification process.

## Experimental setup

Several performance measures are employed in the context of the proposed citrus disease detection system. The metrics under consideration encompass Sensitivity (True Positive Rate), Specificity (True Negative Rate), and Precision (Positive Predictive Value), alongside Accuracy, Area Under the Curve (AUC), and False Positive Rate (FPR), which are presented in Table 2 for further examination and analysis.

## Results and discussions

The results of the proposed approach are evaluated by comparing them with existing methods. The evaluation includes performance metrics, tables, confusion matrices, time complexity

**Table 2. Performance metrics for evaluation.**

| Name | Formula |
|---|---|
| Sensitivity rate | $\frac{TP}{P} = \frac{\text{Number of true positives}}{\text{Number of true positives} + \text{Number of false negatives}}$ |
| Precision rate | $\frac{TP}{TP+FP}$ |
| F1-Score | $2 \times \frac{\text{Precision} \times \text{Recall}}{\text{Precision} + \text{Recall}}$ |
| Accuracy | $\frac{TP+TN}{TP+FP+FN+TN}$ |
| True positive rate (TPR) | $TPR = \frac{\text{True Positive}}{\text{True Positive} + \text{False Negative}}$ |
| False positive rate (FPR) | $FPR = \frac{\text{False Positive}}{\text{False Positive} + \text{True Negative}}$ |

graphs, and the training and testing of the task. Performance metrics are key for evaluating how well the system performs.

## Classification results with DenseNet-201 and AlextNet on hybrid dataset

This experiment used the hybrid dataset for classification with fine-tuned DenseNet-201 deep features. They conducted 10-fold cross-validation with a ratio of 80:20 training to testing. The results are shown in Table 3. The Fine KNN classifier achieved the highest accuracy at 99.5%. You can also find the accuracy details in the confusion matrix presented in Fig 3. The correct classification rates for various diseases in the dataset were as follows: Anthracnose 99.7%, Blackspot 100%, Canker 99.5%, Citrus Scab 98.6%, Greening 100%, and Melanose 99.3%. The Q-SVM classifier achieved the second-best accuracy in this experiment at 99.4%. The remaining classifiers, including C-SVM, Weighted KNN, Linear Discriminant, Fine KNN, Kernel Naive Bayes, Medium Gaussian SVM, Ensemble Subspace Discrimination, Cosine KNN, and Cubic KNN, had accuracies ranging from 99.4%, 99.2%, 99.4%, 91.9%, 99.5%, 99.0%, 99.4%, 99.4%, 97.5, and 98.2% respectively. In the testing phase of this experiment, the processing time was measured. The Fine KNN classifier had a processing time of 23.819 seconds. Importantly, despite its speed, this classifier also achieved superior accuracy compared to other classifiers.

In another experiment, we classified the hybrid dataset using fine-tuned AlexNet deep features. The best accuracy achieved by the Q-SVM classifier was 97.3%. The results and a confusion matrix can be found in Table 4 and Fig 4, respectively. In this dataset, the correct classification rates for different diseases are as follows: Anthracnose 97.7%, Blackspot 99.4%,

**Table 3. Hybrid dataset classification results achieved with fine-tuned DenseNet-201.**

| Classifiers | Recall-Rate (%) | Precision-Rate (%) | False Negative Rate (%) | Area Under Curve | Time (Second) | F1-Score (%) | Accuracy (%) |
|---|---|---|---|---|---|---|---|
| C-SVM | 99.51 | 99.4 | 0.49 | 0.995 | 37.172 | 99.4 | 99.4 |
| W- KNN | 99.31 | 99.2 | 069 | 0.828 | 24.368 | 99.2 | 99.2 |
| Q-SVM | 99.5 | 99.4 | 0.5 | 0.663 | 28.957 | 99.4 | 99.4 |
| LDA | 93.3 | 92.9 | 6.7 | 0.93 | 37.41 | 93.0 | 91.9 |
| F-KNN | 99.5 | 99.5 | 0.5 | 0.99 | 23.819 | 99.5 | 99.5 |
| KNB | 99.1 | 99.0 | 0.9 | 0.99 | 914.83 | 99.0 | 99.0 |
| MG-SVM | 99.5 | 99.4 | 0.5 | 0.995 | 34.896 | 99.4 | 99.4 |
| SDA | 99.4 | 99.45 | 0.6 | 0995 | 287.59 | 99.4 | 99.4 |
| Co-KNN | 97.6 | 97.41 | 2.4 | 0.975 | 25.361 | 97.5 | 97.5 |
| C- KNN | 98.3 | 98.1 | 91.7 | 0.981 | 491.31 | 98.1 | 98.2 |

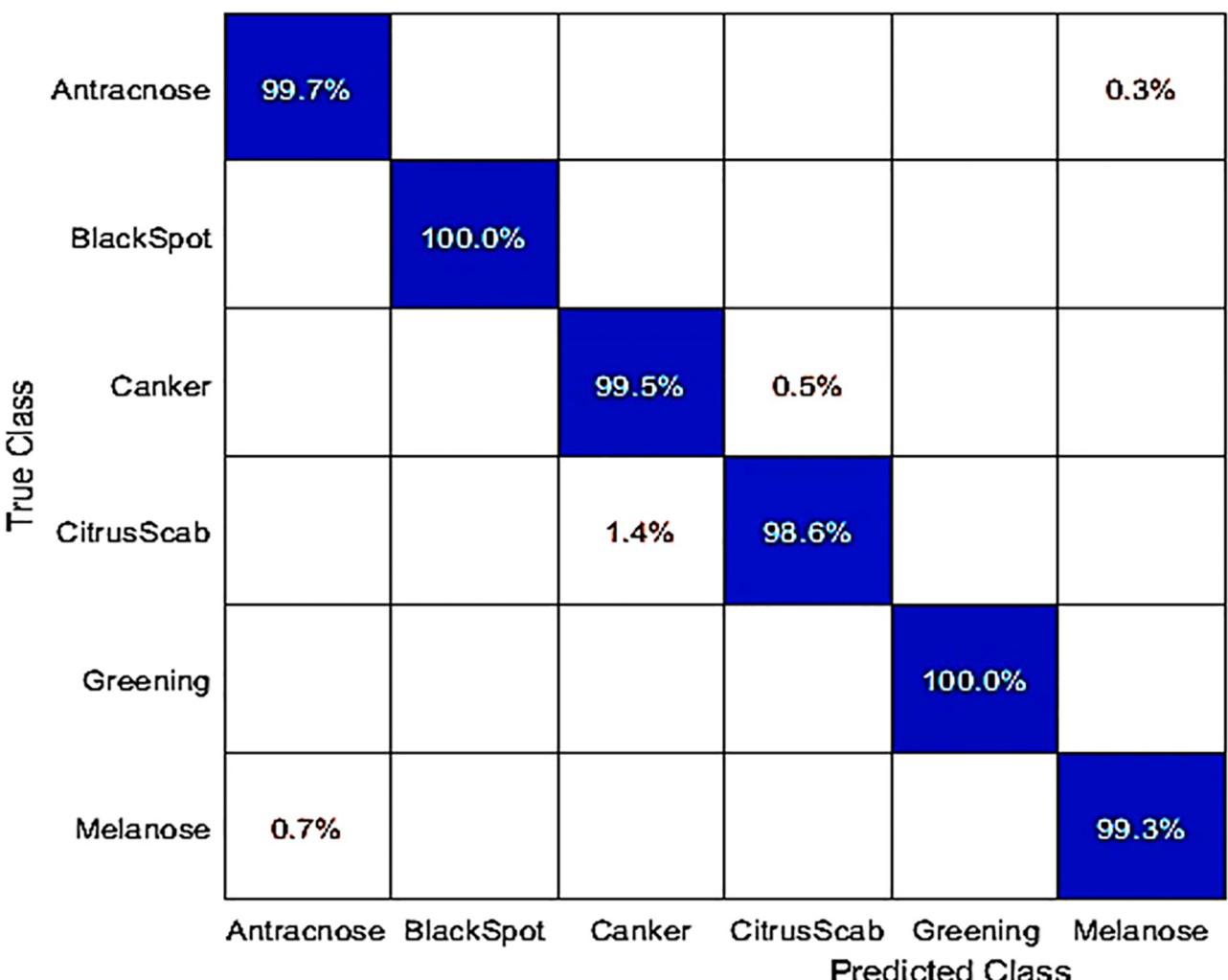

**Fig 3. Confusion matrix for the hybrid dataset, obtained by classifying with the fine-tuned DenseNet-201 using a Fine KNN classifier.**

**Table 4. Hybrid dataset classification results achieved with fine-tuned AlexNet.**

| Classifiers | Recall-Rate (%) | Precision-Rate (%) | False Negative Rate (%) | Area Under Curve | Time (Second) | F1-Score (%) | Accuracy (%) |
|---|---|---|---|---|---|---|---|
| C-SVM | 97.05 | 96.86 | 2.95 | 0.96 | 103.45 | 96.9 | 97.0 |
| W- KNN | 89.6 | 89.1 | 10.4 | 0.89 | 64.095 | 89.3 | 89.5 |
| Q-SVM | 97.3 | 97.15 | 2.7 | 0.97 | 9.0685 | 97.2 | 97.36 |
| LDA | 88.9 | 88.05 | 11.1 | 0.88 | 76.769 | 88.4 | 87.7 |
| F-KNN | 93.7 | 93.8 | 6.3 | 0.93 | 59.533 | 93.7 | 93.9 |
| KNB | 79.2 | 78.05 | 20.8 | 0.78 | 1510.7 | 78.6 | 78.1 |
| MG-SVM | 95.4 | 95.1 | 406 | 0.95 | 115.86 | 95.2 | 95.4 |
| SDA | 83.2 | 83.2 | 16.8 | 0.83 | 1275.7 | 83.2 | 82.8 |
| Co-KNN | 79.1 | 78.8 | 20.9 | 0.79 | 64.854 | 78.9 | 78.9 |
| C- KNN | 75.2 | 74.1 | 24.8 | 0.741 | 1035.1 | 74.6 | 74.1 |

**Model 3**

**Fig 4. Confusion matrix for the hybrid dataset, obtained by classifying with the fine-tuned AlexNet using a Q-SVM classifier.**

Canker 98.2%, Citrus Scab 95.4%, Greening 96.5%, and Melanose 96.6%. The second-best accuracy in this experiment is 97%, achieved by the C-SVM classifier. Other classifiers like Weighted KNN, Q-SVM, Linear Discriminant, Fine KNN, Kernel Navie Bayes, Medium Gaussian SVM, Ensemble Subspace Discrimination, Cosine KNN, and Cubic KNN had accuracies ranging from 97.0%, 89.5%, 97.3%, 87.7%, 93.9%, 78.1%, 95.4%, 82.8%, 78.9%, and 74.1%, respectively. In the testing process, the time taken for the Q-SVM classifier was the lowest at 9.0685 seconds. Interestingly, despite its fast processing time, this classifier also exhibited better accuracy than other classifiers.

In this experiment, we used fine-tuned DenseNet-201 with the best feature selection to classify publicly available hybrid datasets. The results in Table 5 show that Q-SVM achieved the highest accuracy of 99.4%. This accuracy is further detailed in a confusion matrix presented in Fig 5. The correct classification rates for various diseases are quite high in this dataset. For instance, Anthracnose, Blackspot, Canker, Citrus Scab, Greening, and Melanose achieved high accuracy rates of 100%, 100%, 99.3%, 98.4%, 100%, and 99.23% respectively. The second-best accuracy in this experiment was 99.4%, achieved by the ensemble subspace discriminant classifier. The remaining classifiers, including C-SVM, Weighted KNN, Q-SVM, Linear Discriminant, Fine KNN, Kernel Naive Bayes, Medium Gaussian SVM, Ensemble Subspace Discrimination, Cosine KNN, and Cubic KNN, had accuracies ranging from 99.3% 98.9,

**Table 5. Hybrid dataset classification results using the best features of fine-tuned DenseNet-201.**

| Classifiers | Recall-Rate (%) | Precision-Rate (%) | False Negative Rate (%) | Area Under Curve | Time (Second) | F1-Score (%) | Accuracy (%) |
|---|---|---|---|---|---|---|---|
| C-SVM | 99.3 | 99.3 | 0.7 | 0.99 | 14.227 | 99.3 | 99.3 |
| W-KNN | 99.0 | 98.9 | 1 | 0.99 | 11.693 | 98.9 | 98.9 |
| Q-SVM | 99.5 | 99.5 | 0.5 | 0.99 | 13.208 | 99.5 | 99.4 |
| LDA | 99.15 | 99.1 | 0.85 | 0.99 | 9.1895 | 99.1 | 99.1 |
| F-KNN | 99.3 | 99.3 | 0.7 | 0.99 | 11.685 | 99.35 | 99.2 |
| KNB | 98.8 | 98.8 | 1.2 | 0.98 | 308.63 | 98.8 | 98.7 |
| MG-SVM | 99.4 | 99.4 | 0.6 | 0.99 | 15.708 | 99.4 | 99.3 |
| SDA | 99.5 | 99.4 | 0.5 | 0.99 | 59.336 | 99.4 | 99.4 |
| Co-KNN | 98.05 | 97.8 | 1.95 | 0.94 | 11.688 | 97.1 | 97.9 |
| C- KNN | 98.25 | 98.0 | 1.75 | 0.98 | 218.7 | 98.1 | 98.1 |

99.4% 99.1% 99.2% 98.7% 99.3% 99.4% 97.4% 97.9% and 98.1%, respectively. In the testing phase, the time taken for processing is also recorded in this experiment. The Q-SVM classifier exhibited a processing time, which was 13.208 seconds. Compared to the results in Table 3, this experiment demonstrated improved computational time and accuracy performance.

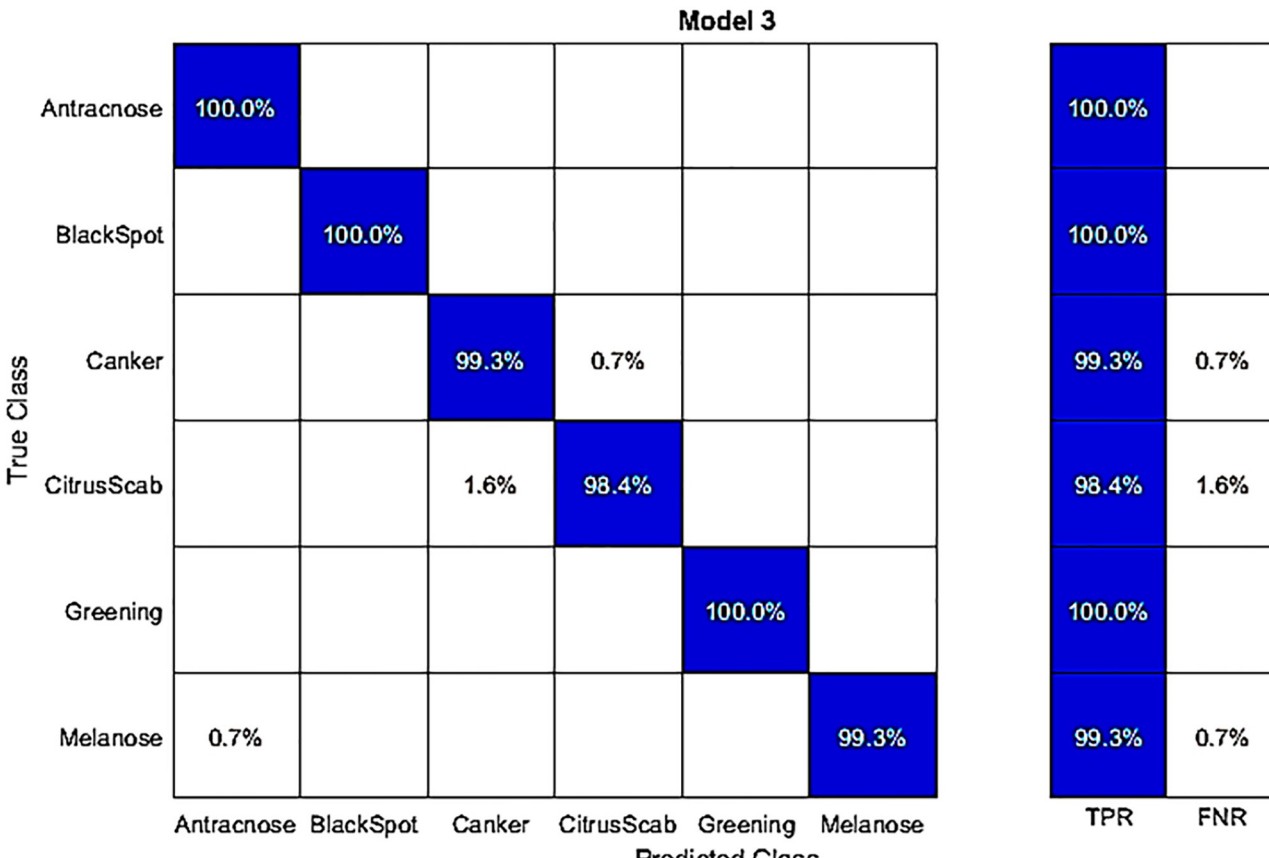

**Fig 5. Confusion matrix for the hybrid dataset on best features, obtained by classifying with the fine-tuned DenseNet-201 using a Q-SVM classifier.**

## Results of Leaves dataset

In this experiment, we applied classification to the publicly available Leaves dataset using fine-tuned DenseNet-201 deep features. The results of this classification are displayed in Table 6. Among the classifiers, Q-SVM achieved the highest accuracy of 98.9%, which is also visualized in the confusion matrix shown in Fig 6. In this dataset, the accurate classification rates for Blackspot, Melanose, Canker, Greening, and Healthy are 98.2%, 100%, 100%, 97.1%, and 100%, respectively. The second-best accuracy in this experiment is 98.9%, achieved by the C-SVM classifier. Other classifiers, including Weighted KNN, Linear Discriminant, Fine KNN, Kernel Naive Bayes, Medium Gaussian SVM, Ensemble Subspace Discrimination, Cosine KNN, and Cubic KNN, have accuracies ranging from 98% to 97.1%. The processing time was measured in the testing phase, with the Q-SVM classifier recording the processing time at 13.294 seconds. Additionally, this classifier exhibited superior accuracy compared to the others.

This experiment's classification was conducted on the Leaves dataset using fine-tuned Alex-Net deep features. The results are shown in Table 7, with Q-SVM achieving the highest accuracy of 94.3%. This accuracy is also reflected in the associated confusion matrix, presented in Fig 7. In this dataset, the accurate classification rates for Blackspot, Melanose, Canker, Greening, and Healthy are 91.8%, 100%, 99.4%, 85.8%, and 97.4%, respectively. The second-highest accuracy in this experiment is 94.2%, achieved by the C-SVM classifier. The remaining classifiers have accuracies ranging from 94.3% to 87.3%. Processing time during testing was measured in this experiment, with the Q-SVM classifier having the observed time at 38.894 seconds. Notably, the accuracy of this classifier outperformed the others.

## Results of Fruits dataset

In this experiment, we conducted classification on the publicly available Fruit dataset using fine-tuned DenseNet-201 deep features. The results are displayed in Table 8, and the Weighted KNN classifier attained the highest accuracy of 99.4%, which is also corroborated by a confusion matrix illustrated in Fig 8. In this dataset, the accurate classification rates for Blackspot, Canker, Greening, Scab, and Healthy are 100.0%, 99.2%, 100.0%, 96.7%, and 100.0%, respectively. The second-highest accuracy in this experiment is 99.4% achieved by the Linear Discriminant classifier. The remaining classifiers, such as C-SVM, Q-SVM, Linear Discriminant, Fine KNN, Kernel Navie Bayes, Medium Gaussian SVM, Ensemble Subspace Discrimination, Cosine KNN, and Cubic KNN, have accuracies of 99.2%, 99.4%, 99.2%, 99.4%, 98.2%, 97.0%,

**Table 6. Leaves dataset classification results achieved with fine-tuned DenseNet-201.**

| Classifiers | Recall-Rate (%) | Precision-Rate (%) | False Negative Rate (%) | Area Under Curve | Time (Second) | F1-Score (%) | Accuracy (%) |
|---|---|---|---|---|---|---|---|
| C-SVM | 99.06 | 99.06 | 0.94 | 0.99 | 13.763 | 99.06 | 98.9 |
| W-KNN | 97.96 | 98.4 | 2.04 | 0.982 | 9.7902 | 98.17 | 98.0 |
| Q-SVM | 99.06 | 99.06 | 0.94 | 0.99 | 13.294 | 99.06 | 98.9 |
| LDA | 98.46 | 98.58 | 1.54 | 0.986 | 13.451 | 98.51 | 98.3 |
| F-KNN | 98.3 | 98.1 | 1.7 | 0.982 | 9.6864 | 98.1 | 97.9 |
| KNB | 97.72 | 97.96 | 2.28 | 0.98 | 319.78 | 97.83 | 97.4 |
| MG-SVM | 98.6 | 98.64 | 1.4 | 0.986 | 15.226 | 98.6 | 98.4 |
| SDA | 60.74 | 67.6 | 39.26 | 0.678 | 145.63 | 63.9 | 63.1 |
| Co-KNN | 98.12 | 98.44 | 1.88 | 0.986 | 11.216 | 98.2 | 98.0 |
| C-KNN | 97.74 | 98.22 | 2.26 | 0.982 | 141.23 | 97.9 | 97.7 |

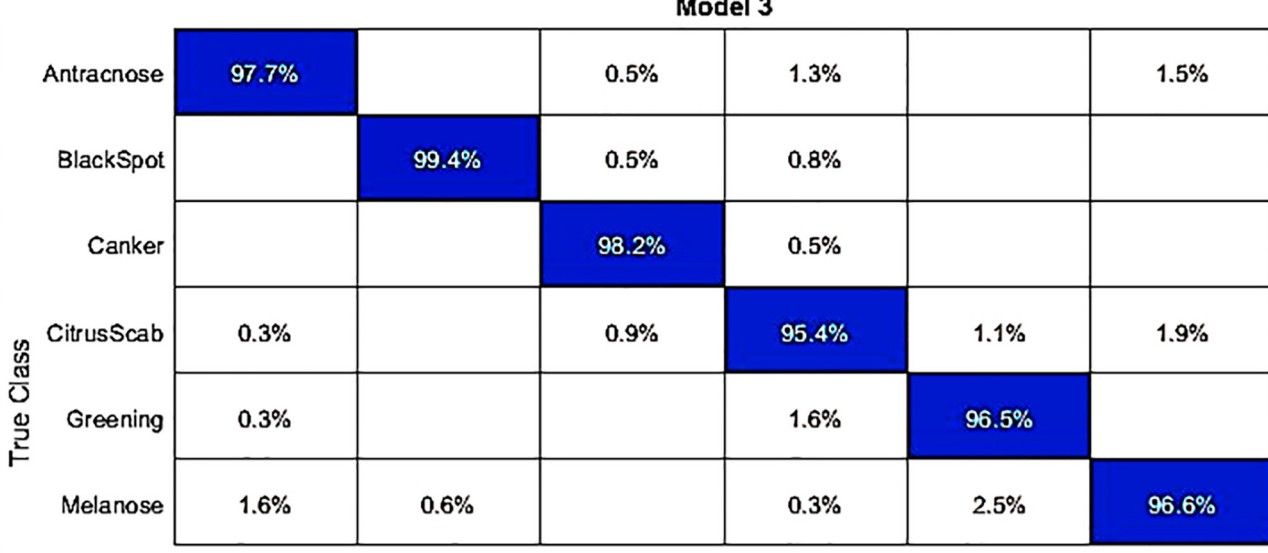

**Fig 6. Confusion matrix for the Leaves dataset using fine-tuned DenseNet-201 deep features.**

98.9%, 92.3%, 94.9%, and 93.8%, respectively. During the testing phase, the time taken for processing is measured in this experiment. The Linear Discriminant classifier had a processing time of 6.3224 seconds, and it also exhibited superior accuracy compared to other classifiers.

In this experiment, we conducted classification using publicly available datasets known as the Fruit dataset, employing fine-tuned AlexNet deep features. The results of this classification are displayed in Table 9. Among the classifiers, Q-SVM achieved the highest accuracy of

**Table 7. Leaves dataset classification results achieved with fine-tuned AlexNet.**

| Classifiers | Recall-Rate (%) | Precision-Rate (%) | False Negative Rate (%) | Area Under Curve | Time (Second) | F1-Score (%) | Accuracy (%) |
|---|---|---|---|---|---|---|---|
| C-SVM | 95.24 | 94.86 | 4.76 | 0.948 | 43.431 | 95.04 | 94.2 |
| W-KNN | 89.58 | 89 | 10.42 | 0.89 | 24.928 | 89.2 | 88.7 |
| Q-SVM | 95.24 | 94.9 | 4.76 | 0.948 | 38.894 | 95.06 | 94.3 |
| LDA | 93.5 | 93.62 | 6.5 | 0.934 | 39.553 | 93.5 | 92.6 |
| F-KNN | 89.04 | 89.98 | 10.96 | 0.9 | 26.844 | 89.5 | 87.4 |
| KNB | 89.76 | 89.36 | 10.24 | 0.89 | 776.3 | 89.5 | 88.0 |
| MG-SVM | 94.46 | 93.64 | 5.54 | 0.792 | 41.522 | 94.0 | 93.2 |
| SDA | 54.46 | 59.98 | 45.54 | 0.6 | 351.78 | 57.1 | 54.3 |
| Co-KNN | 90.98 | 89.86 | 9.02 | 0.896 | 30.436 | 90.4 | 89.7 |
| C-KNN | 88.74 | 87.3 | 11.26 | 0.874 | 362.36 | 88.0 | 88.1 |

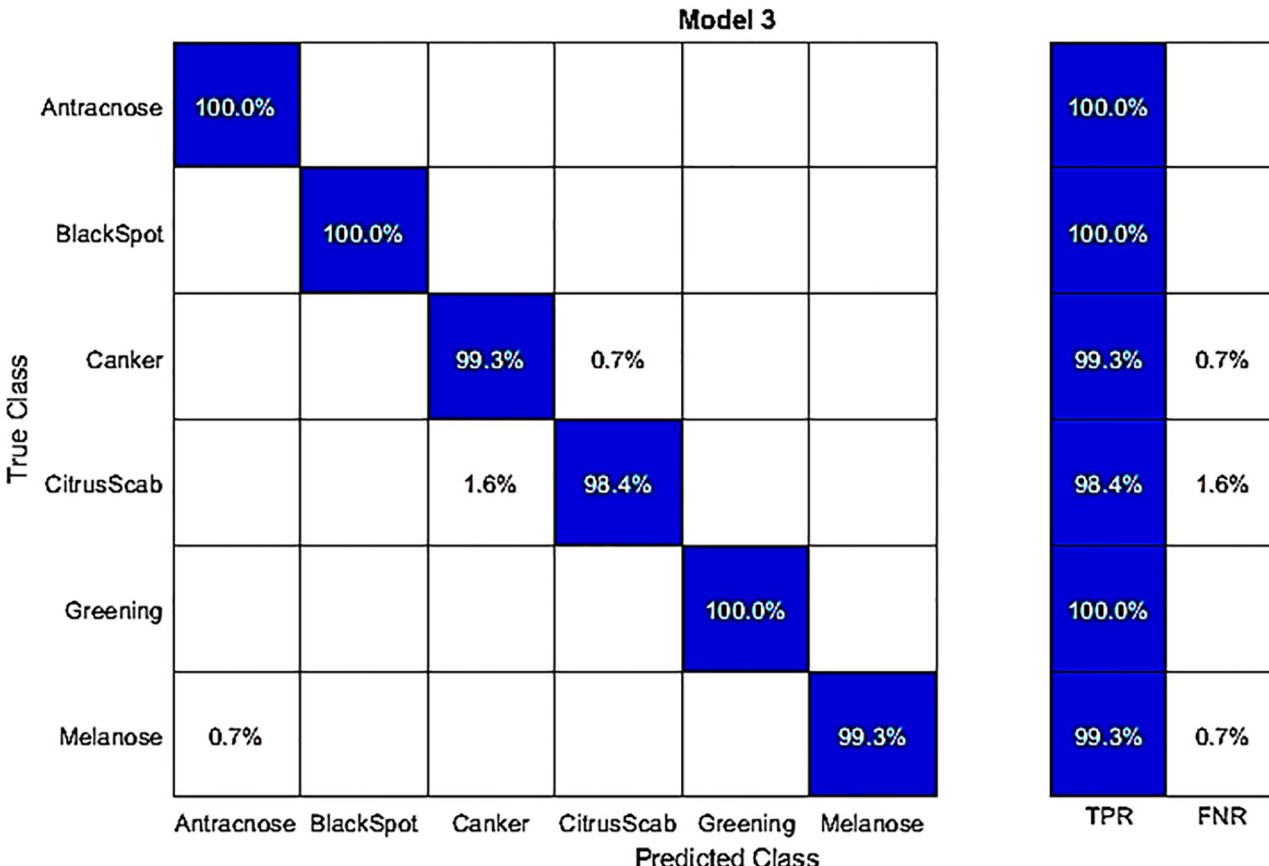

**Fig 7. Confusion matrix for the Leaves dataset using fine-tuned AlexNet deep features.**

93.7%, also depicted in a confusion matrix shown in Fig 9. This dataset's accurate classification rates for categories like Blackspot, Canker, Greening, Scab, and Healthy are 86.8%, 93.8%, 99.2%, 90.0%, and 92.4%, respectively. The second-best accuracy in this experiment is 92.5%, achieved by the C-SVM classifier. As for the remaining classifiers, such as C-SVM, Weighted KNN, Q-SVM, Linear Discriminant, Fine KNN, Kernel Navie Bayes, Medium Gaussian SVM, Ensemble Subspace Discrimination, Cosine KNN, and Cubic KNN, their accuracies are 91.3%,

**Table 8. Fruits dataset classification results achieved with fine-tuned DenseNet-201.**

| Classifiers | Recall Rate (%) | Precision Rate (%) | False Negative Rate (%) | Area Under Curve | Time (Second) | F1-Score (%) | Accuracy (%) |
|---|---|---|---|---|---|---|---|
| C-SVM | 98.58 | 99.24 | 1.42 | 0.986 | 12.122 | 98.90 | 99.2 |
| W- KNN | 97.18 | 96.36 | 2.82 | 0.97 | 4.646 | 96.76 | 99.4 |
| Q-SVM | 98.58 | 99.24 | 1.42 | 1 | 9.320 | 98.90 | 99.2 |
| LDA | 99.4 | 80.033 | 0.6 | 0.992 | 6.3224 | 88.67 | 99.4 |
| F-KNN | 98.02 | 97.74 | 1.98 | 0.097 | 3.953 | 97.87 | 98.2 |
| KNB | 97.32 | 96.72 | 2.68 | 0.966 | 411.7 | 97.01 | 97.0 |
| MG-SVM | 99.08 | 98.14 | 0.92 | 0.98 | 10.329 | 98.6 | 98.9 |
| SDA | 89.84 | 92.4 | 10.16 | 0.924 | 51.123 | 91.10 | 92.3 |
| Co-KNN | 95.16 | 96.16 | 4.84 | 0.96 | 4.298 | 95.6 | 94.9 |
| C- KNN | 94.6 | 95.4 | 5.4 | 0.95 | 45.423 | 94.9 | 93.8 |

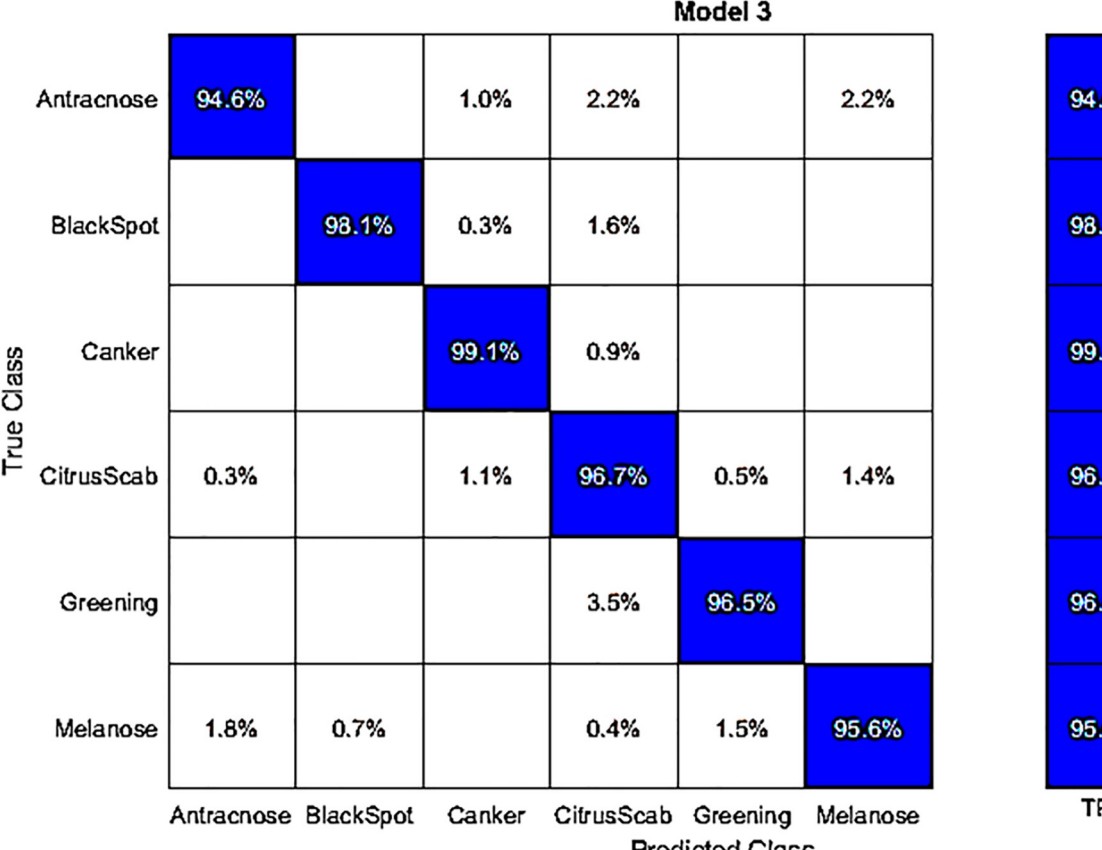

**Fig 8. Confusion matrix for the Fruits dataset using fine-tuned DenseNet-201 deep features.**

71.3%, 93.7%, 92.5%, 8%, 85.4%, 71.5%, and 75.7%, respectively. In this experiment, the processing time during the testing phase is measured. The Q-SVM classifier has a processing time of 30.932 seconds. It is worth noting that the accuracy of this classifier is superior to other classifiers. Additionally, we experimented by using fine-tuned DenseNet-201 with the best features to classify the Fruit dataset. The results in Table 10 show that C-SVM achieved the highest accuracy of 99.2% with a time of 2.9 seconds.

**Table 9. Fruits dataset classification results achieved with fine-tuned AlexNet.**

| Classifiers | Recall Rate (%) | Precision Rate (%) | False Negative Rate (%) | Area Under Curve | Time (Second) | F1-Score (%) | Accuracy (%) |
|---|---|---|---|---|---|---|---|
| LDA | 88.78 | 89.32 | 11.22 | 0.894 | 26.456 | 89.0 | 91.3 |
| KNB | 71.54 | 77.6 | 28.46 | 0.776 | 675.16 | 74.4 | 71.3 |
| Q-SVM | 94.7 | 92.88 | 5.3 | 0.928 | 30.932 | 93.7 | 93.7 |
| C-SVM | 93.1 | 91.62 | 6.9 | 0.916 | 31.286 | 92.3 | 92.5 |
| MG-SVM | 80.1 | 88.24 | 19.9 | 0.88 | 26.864 | 83.9 | 89.7 |
| F-KNN | 87.14 | 90.14 | 12.86 | 0.9 | 16.291 | 88.6 | 88.5 |
| Co- KNN | 79.12 | 80.2 | 20.88 | 0.802 | 17.797 | 79.6 | 80.4 |
| W- KNN | 84.0 | 84.76 | 16 | 0.846 | 16.424 | 84.3 | 85.4 |
| SDA | 67.12 | 69.7 | 32.88 | 0.698 | 162.84 | 68.3 | 71.5 |
| C-KNN | 77.04 | 74.76 | 22.96 | 0.748 | 129.79 | 75.8 | 75.7 |

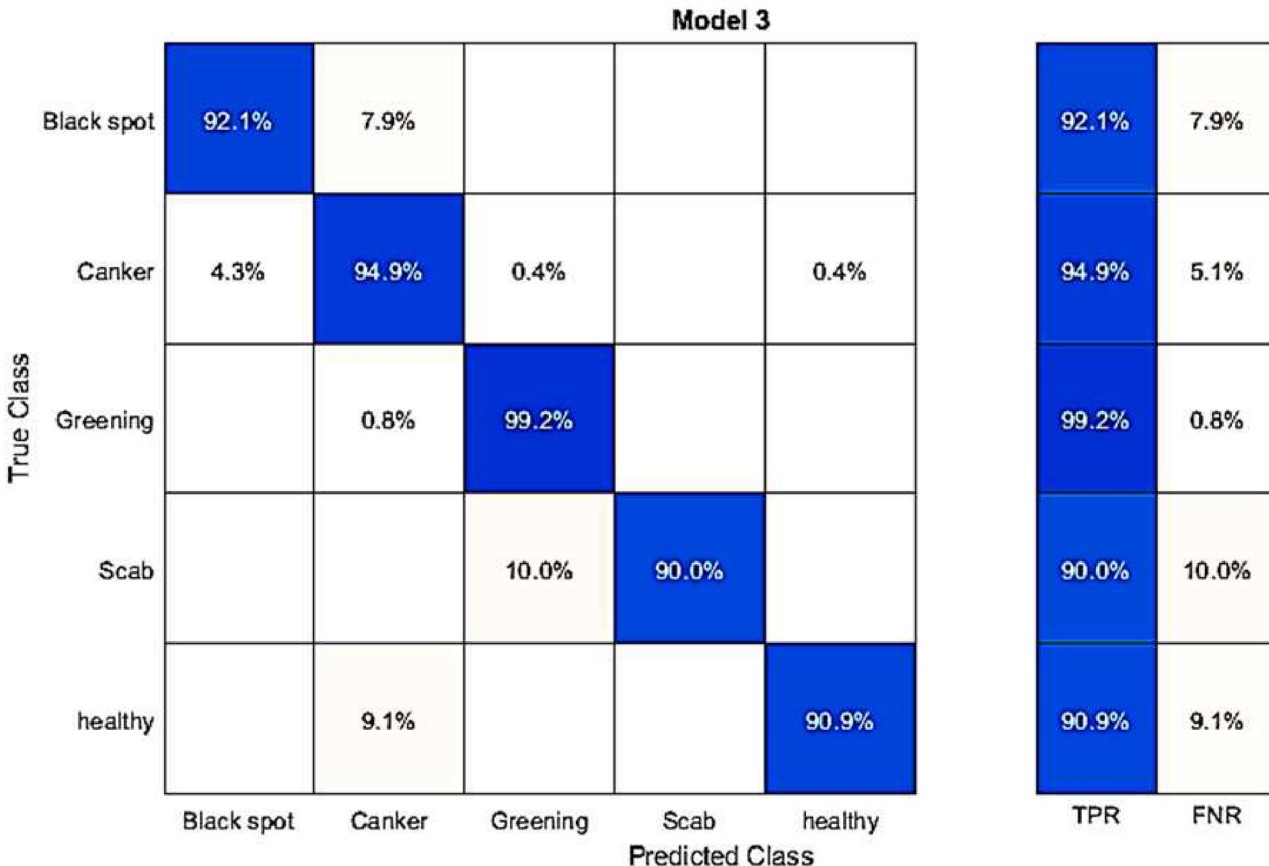

**Fig 9. Confusion matrix for the Fruits dataset using fine-tuned AlexNet deep features.**

## Computational complexity analysis

We have performed the computational complexity analysis. Time measurements are recorded using a hybrid dataset, as depicted in Fig 10. Furthermore, the time aspect using leaves data is considered, as depicted in Fig 11. Furthermore, time-related fruit data is recorded and presented in Fig 12.

**Table 10. Fruits dataset classification results using the best features of fine-tuned DenseNet-201.**

| Classifiers | Recall Rate (%) | Precision Rate (%) | False Negative Rate (%) | Area Under Curve | Time (Second) | F1-Score (%) | Accuracy (%) |
|---|---|---|---|---|---|---|---|
| C-SVM | 97.1 | 96.3 | 2.9 | 0.97 | 2.9 | 96.76 | 99.2 |
| W- KNN | 97.0 | 97.02 | 3.0 | 0.97 | 30.1 | 97.5 | 97.0 |
| Q-SVM | 99.6 | 99.2 | 0.4 | 1.0 | 7.6 | 99.4 | 99.1 |
| LDA | 98.5 | 98.4 | 1.5 | 0.985 | 17.7 | 98.4 | 98.5 |
| F-KNN | 98.3 | 99.0 | 1.7 | 0.983 | 19.7 | 98.5 | 98.3 |
| KNB | 95.6 | 95.0 | 4.4 | 0.956 | 16.7 | 95.3 | 95.5 |
| MG-SVM | 98.8 | 99.4 | 1.2 | 0.988 | 5.7 | 99.1 | 98.8 |
| SDA | 95.3 | 96.0 | 4.7 | 0.953 | 31.7 | 95.5 | 95.5 |
| Co- KNN | 98.9 | 98.8 | 1.1 | 0.988 | 29.4 | 98.9 | 98.9 |
| C- KNN | 95.5 | 95.6 | 4.5 | 0.956 | 9.7 | 95.6 | 95.5 |

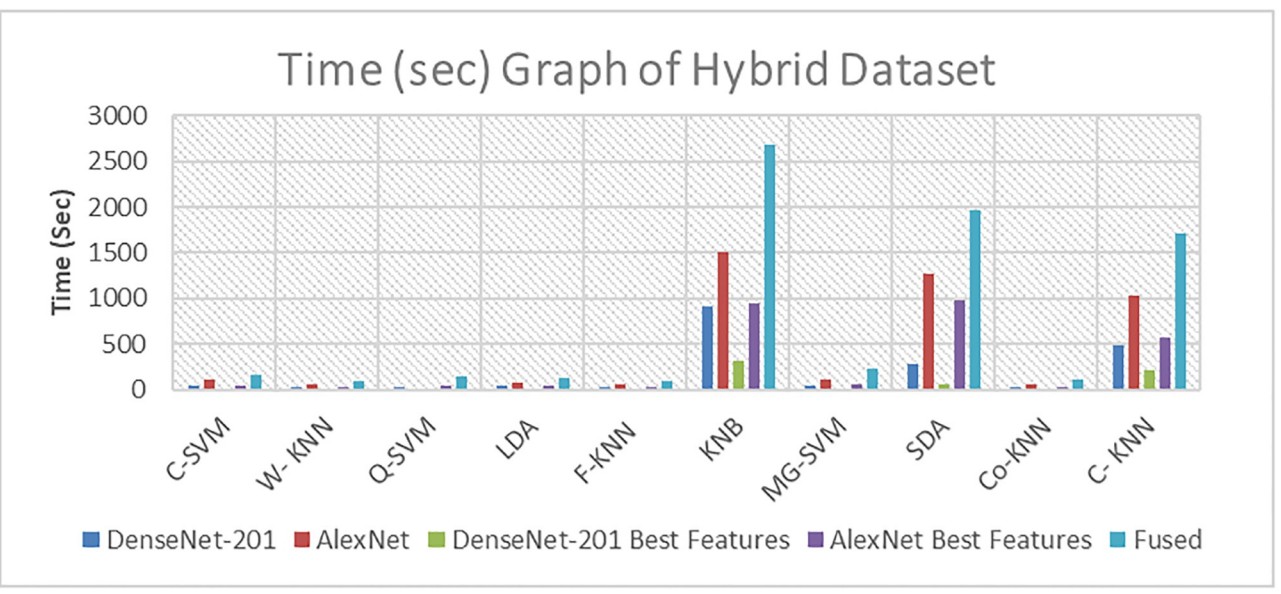

**Fig 10. Time (in second) graph of the hybrid dataset.**

## Discussions

This section offers a comprehensive discussion of the proposed framework, presenting results across Tables 3 to 10. The experiments encompass three datasets, with five experiments conducted for each dataset. The results for the hybrid dataset are found in Tables 3–5. They reveal the highest accuracy values to be 99.5%, 97.36%, and 99.4%, respectively. The accuracy analysis shows that applying the feature selection method to DenseNet-201 features slightly decreases accuracy from 99.5% to 99.4%. Likewise, for AlexNet features, the accuracy decreases from 97.36% to 96.9% when the feature selection method is applied. A fusion step is executed, and

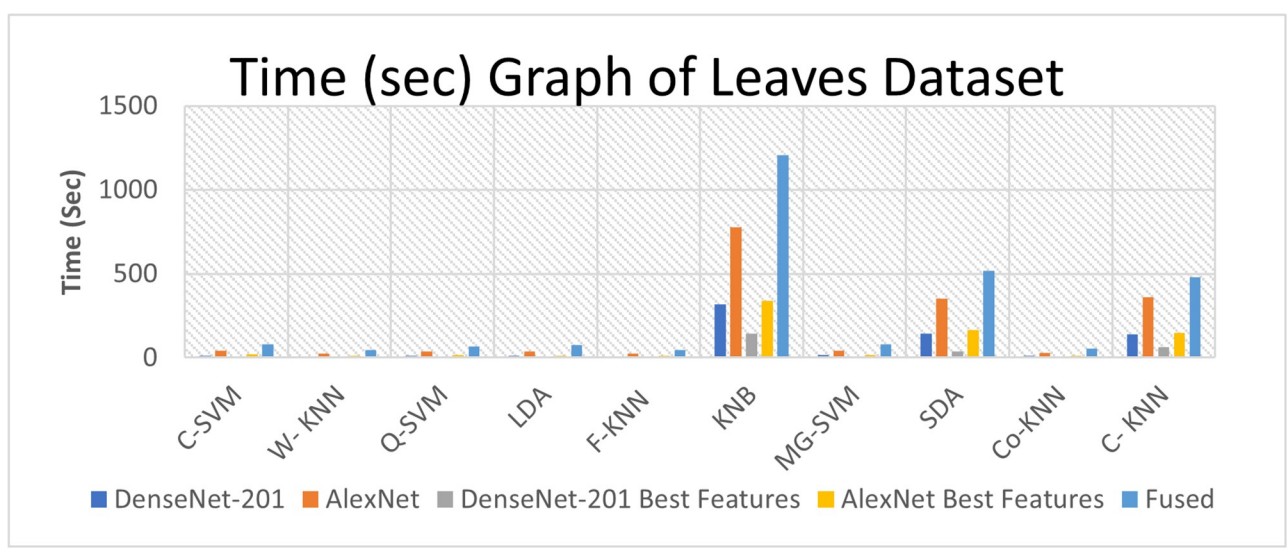

**Fig 11. Time (in second) graph of leaves dataset.**

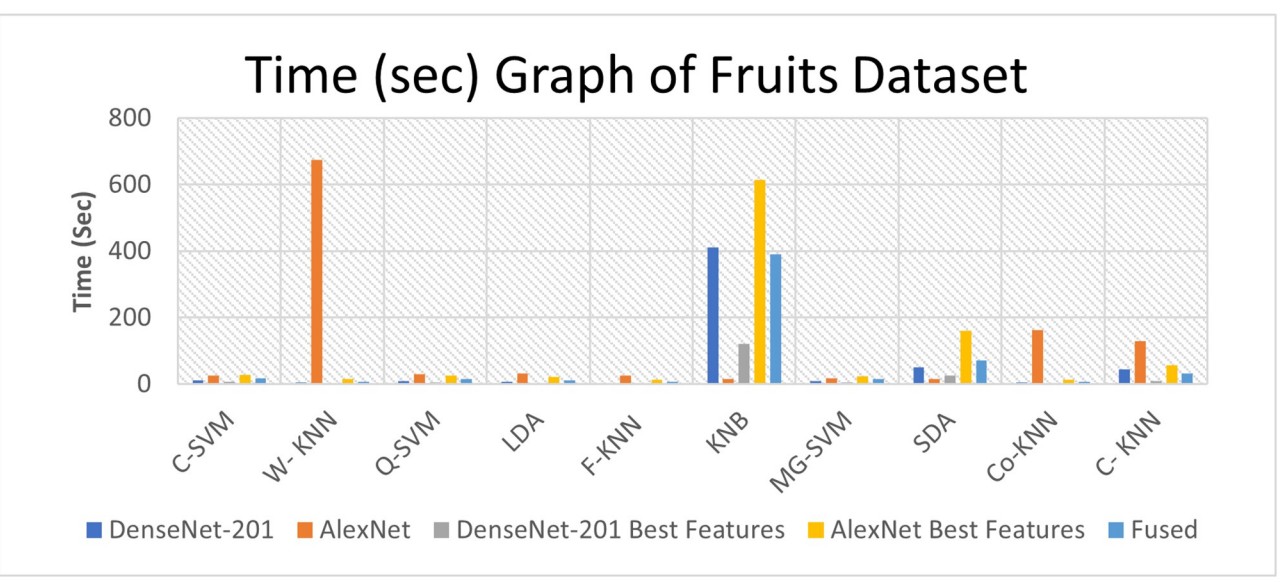

**Fig 12. Time (in second) graph of fruits dataset.**

the results are presented in Table 5. Notably, the accuracy improves, but there is an increase in computational time.

In the leaves dataset, the results in Tables 6, and 7 indicate the highest accuracy values of 98.9%, and 94.3%, achieved by the C-SVM and Q-SVM, respectively. Similarly, the results of the fruit dataset presented in Tables 8–10, indicate the highest accuracy values of 99.4%, 93.7%, and 99.2%, achieved by Weighted KNN, Q-SVM, and C-SVM, respectively. When the best features are applied for both hybrid and fruit datasets, it reveals that the accuracy decreased slightly from 99.5% to 99.4%, and 99.4% to 99.2%, while the time is slightly reduced from 23.819 to 9.1895 seconds, and 3.953 to 2.9 seconds, respectively.

The application of feature selection and fusion methods, combined with deep learning models for citrus disease identification, has shown considerable improvements over individual classifiers proposed in the literature. Experiments were conducted on three datasets—Hybrid, Leaves, and Fruit—with five experiments on each. This section critically analyzes the results, compares them with existing literature, and discusses their implications for future research.

Challenges and future research directions increased computational time after feature fusion, especially in the Hybrid and Leaves datasets, highlights the challenge of deploying deep learning models [27–29] for real-time agricultural applications. Future work should focus on optimizing computational efficiency through techniques like model pruning, quantization, or distillation to reduce the complexity of fused models without sacrificing much accuracy, enabling deployment on resource-constrained devices like mobile phones.

## State of the art comparisons

A comparison of the results of the proposed framework to the recent state-of-the-art techniques is given in Table 11. Comparison with State-of-the-Art Methods compares the proposed framework to existing methods. Our approach demonstrated superior performance, achieving 99.9% and 99.6% accuracies with DenseNet-201 for the Hybrid and Leaves datasets.

**Table 11. Comparisons with existing techniques.**

| Reference | Year | Accuracy (%) |
|---|---|---|
| Chen X. et al, [30] | 2021 Fruits Dataset | 99% |
| Ahmed Elaraby, et al, [31] | 2022 Fruits Dataset | 94% |
| Abdul Majid, et al, [32] | 2022 Fruits Dataset | 99% |
| Poonam Dhiman, et al, [33] | 2023 Fruits Dataset, Leaves Dataset | 97.1% & 98.2% |
| Vinay Kukreja, et al, [34] | 2023 Leaves Dataset | 94.0% |
| Proposed Model | Hybrid Dataset (Fruits and Leaves) | 99.9% |
| | Leaves Dataset | 99.6% |
| | Fruits Dataset | 99.1% |

## Conclusion and future direction

This study investigates deep learning and has made remarkable strides in agriculture, notably in its ability to automate the identification of plant diseases while reducing the need for extensive human involvement. It aimed to create an automated system for identifying diseases in Citrus fruits and leaves using deep learning approaches and best-feature selection. Data augmentation is performed early to expand the dataset and enhance the robustness of deep learning models. Subsequently, two pre-trained models were adapted and fine-tuned using transfer learning on the augmented datasets. Features were extracted through a proposed MFO method and combined using a serial-based approach. The resulting features were then classified using various supervised learning algorithms. The experimental process covered three distinct datasets: Hybrid dataset, Leaves dataset, and Fruits dataset, achieving high accuracies of 99.9%, 99.6%, and 99.1%, respectively. This study showcases the efficacy of deep learning and feature selection in automating the identification of citrus fruit and leaf diseases with remarkable accuracy. In the future, this research expand the scope to include additional datasets with diverse and complex images encompassing various diseases. Moreover, it should also focus on improving the computational efficiency of the model for real-time use in field conditions, exploring model compression techniques, and enhancing generalization by incorporating diverse environmental data to address potential dataset biases.

## Author Contributions

**Conceptualization:** Nouman Butt, Norma Latif Fitriyani, Yeonghyeon Gu, Muhammad Syafrudin.

**Data curation:** Nouman Butt, Ali Raza.

**Formal analysis:** Nouman Butt, Muhammad Munwar Iqbal, Shabana Ramzan, Laith Abualigah, Norma Latif Fitriyani.

**Funding acquisition:** Yeonghyeon Gu, Muhammad Syafrudin.

**Investigation:** Muhammad Munwar Iqbal, Shabana Ramzan, Ali Raza, Laith Abualigah.

**Methodology:** Muhammad Syafrudin.

**Project administration:** Muhammad Munwar Iqbal, Muhammad Syafrudin.

**Resources:** Muhammad Munwar Iqbal, Shabana Ramzan, Laith Abualigah.

**Software:** Ali Raza.

**Supervision:** Yeonghyeon Gu, Muhammad Syafrudin.

**Validation:** Shabana Ramzan, Ali Raza, Laith Abualigah, Norma Latif Fitriyani.

**Visualization:** Norma Latif Fitriyani.

**Writing – original draft:** Nouman Butt, Muhammad Munwar Iqbal, Shabana Ramzan, Ali Raza.

**Writing – review & editing:** Norma Latif Fitriyani, Yeonghyeon Gu, Muhammad Syafrudin.

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
