## [Decision Letter · Decision Letter 0]

12 Sep 2024

PONE-D-24-36174Citrus Diseases Detection Using Innovative Deep Learning Approach and Hybrid Meta-HeuristicPLOS ONE

Dear Dr. Syafrudin,

Thank you for submitting your manuscript to PLOS ONE. After careful consideration, we feel that it has merit but does not fully meet PLOS ONE’s publication criteria as it currently stands. Therefore, we invite you to submit a revised version of the manuscript that addresses the points raised during the review process.

We have received feedback from our reviewers, and after careful consideration, we have decided to request major revisions. The reviewers provided valuable insights and suggestions, highlighting areas that require further development. We encourage you to address their comments thoroughly to enhance the quality of your work.

Additionally, the reviewers suggested several references that may be beneficial to your manuscript. Please consider incorporating these references in the revised version, provided they align with the scope and objectives of your study.

We look forward to receiving your revised manuscript.

We look forward to receiving your revised manuscript.

Kind regards,

Hirenkumar Kantilal Mewada

Academic Editor

PLOS ONE

2. Please note that PLOS ONE has spec6ific guidelines on code sharing for submissions in which author-generated code underpins the findings in the manuscript. In these cases, all author-generated code must be made available without restrictions upon publication of the work. Please review our guidelines at https://journals.plos.org/plosone/s/materials-and-software-sharing#loc-sharing-code and ensure that your code is shared in a way that follows best practice and facilitates reproducibility and reuse.

3. Thank you for stating the following financial disclosure: [This work was supported by Institute of Information & communications Technology Planning & Evaluation (IITP) grant funded by the Korea government(MSIT) (No.1711160571, MLOps Platform for Machine learning pipeline automation.)]. Please state what role the funders took in the study. If the funders had no role, please state: "The funders had no role in study design, data collection and analysis, decision to publish, or preparation of the manuscript." If this statement is not correct you must amend it as needed. Please include this amended Role of Funder statement in your cover letter; we will change the online submission form on your behalf.

Reviewers' comments:

Reviewer's Responses to Questions

**Comments to the Author**

1. Is the manuscript technically sound, and do the data support the conclusions?

Reviewer #1: Yes

Reviewer #2: Partly

2. Has the statistical analysis been performed appropriately and rigorously? 

Reviewer #1: Yes

Reviewer #2: No

3. Have the authors made all data underlying the findings in their manuscript fully available?

Reviewer #1: Yes

Reviewer #2: No

4. Is the manuscript presented in an intelligible fashion and written in standard English?

Reviewer #1: No

Reviewer #2: No

5. Review Comments to the Author

Reviewer #1: The study's topic is attractive; however, it requires significant modifications. The authors need to revise specific sections with greater attention.

- Lines (47-48), the full names of all models should be provided upon their first mention.

- Lines (35, 37), CV should be written out in full initially, with no need to repeat the Literature Review.

- What is the difference between 'CV' mentioned in the introduction and 'convolutional neural networks (CV)' in line 103? Please verify.

- Figure 1 needs to be zoomed in to clearly display its details.

- Please insert numbering for the manuscript headings, according to the journal guidelines

Proposed Methodology, Data Augmentation, Deep Feature Extraction, Modified DenseNet201…………… etc.

- Table 1. Procedure for constructing flames in the optimization, and then “Procedure for constructing flames in the optimization:” Do not repeat the rewrite again?

- The authors mentioned 'Results and Discussions' before 'Experimental Setup,' which is not appropriate. Please move the 'Experimental Setup' section to 'Materials and Methods.'

- Table 2, there is no need to repeat the symbols again, such as 'F1-Score' and 'Accuracy'.

- “Results on Hybrid Dataset”, revise the heading to ensure it aligns with this section.

- Figures 3-9 have low resolution; please improve it.

- Table 5: Why isn't the same style used as in Tables 6, 7, and 8? Please add % to the first row and remove it from the other fields—there's no need to repeat it. Please fix this issue throughout the manuscript.

- The manuscript contains too many confusion matrices. Limit them to two and include the learning curve for the superior model to emphasize the novelty.

- The discussion in the paper is weak; please include more previous studies, according to the lasted studies

- https://doi.org/10.1016/j.dib.2024.110713

- https://doi.org/10.3390/plants13010135

- https://doi.org/10.1007/s11760-024-03452-2

- https://doi.org/10.1016/j.compag.2024.108907

- https://doi.org/10.1016/j.compag.2022.107453

- Line (362), Here are the potential areas for future work to…….., Please rewrite this sentence, as it is not ideal to start with 'Here'.

- This paper requires significant revisions before it can be considered complete. Therefore, I recommend a major revision. Thank you.

Reviewer #2: Reviewers Comments

A major revision is being suggested. Following are the point-to-point comments that need to be addressed for further review:

1. Abstract: The motivation of the research is not clear. It is suggested that the authors include a clear motivation statement in the abstract. Also, one sentence presenting the major findings with quantitative support needs to be included as the second sentence from the end of the abstract.

2. Introduction: This is written very poorly. The authors are suggested to merge the Literature review with the introduction to correctly identify the research gap and formulate the objectives in line with the research gap.

3. Line 130: The authors have mentioned that this study utilizes three datasets. However, no details related to the data set are provided. Also, the author has not declared that the dataset is public or private, developed by authors, or accessed from other sources. The details must be stated, and appropriate citations must be provided.

4. Figure 3, Figure 6, figure 7, figure 8, figure 9: The quality of the figure must be enhanced. The current figure quality is blurry and difficult to read.

5. Figure 10, Figure 11, and Figure 12 present the same quantitative values throughout the graph. Hence, it is suggested that the authors remove the table below and make good-quality plots for better representation and understanding.

6. Line 319 to 347, the Authors discussed the obtained results, and one table has been appended below with the title State of the art comparisons. However, the critical comparison and reasoning behind the obtained results are missing in the discussion section. The authors are expected to discuss the obtained results and compare them with the existing results to explain the reasoning behind the presented phenomenon. Hence, it was suggested that the discussion be enhanced as suggested.

7. The conclusions must be proper and in alignment with the objectives. Also, it is expected that authors must present major findings with quantitative support. The authors obtained multiple parameters but only presented parameter accuracy and provided it for the three datasets. No proper and critical write-up is expected here. Hence, the authors are advised to rewrite the conclusion in a maximum of 200 words and include 1-2 sentences at the end that relate to the future direction of the work.

8. References: The authors are advised to include more recent references, particularly from journals such as Elsevier, Springer, PLOS ONE, etc. Currently, no articles from 2024 have been cited. Only two articles from 2023 and one from 2022 are referenced. This suggests either an inadequate literature survey or an outdated research domain. It is strongly recommended that the authors incorporate recent citations to ensure the study's relevance.

6. PLOS authors have the option to publish the peer review history of their article (what does this mean?). If published, this will include your full peer review and any attached files.

Reviewer #1: No

Reviewer #2: **Yes: **Dr. Narendra Khatri

---

## [Author Response · Author response to Decision Letter 0]

12 Nov 2024

Response to the Reviewer 1

We are thankful to the reviewer for the encouraging and positive comments to improve the manuscript. We have revised our manuscript based on reviewer 1’s comments and highlighted them in the revised manuscript. Please, find below the referees’ comments and our responses inserted after each comment in italics.

Review Comments to the Author

Reviewer #1

The study's topic is attractive; however, it requires significant modifications. The authors need to revise specific sections with greater attention.

Author response: We sincerely thank the reviewer for the encouraging and valuable comments. We are sorry for the mistake caused by our negligence. In the revised manuscript, the mistake has been corrected and highlighted. We have tried our best to address all the concerns and comments and please find our point-by-point response in the following. We hope our amendments are satisfactory and consider our current version of the manuscript for publication.

Reviewer #1, Concern #1: Lines (47-48), the full names of all models should be provided upon their first mention.

Author response: Thank you very much for your insightful and constructive feedback. We sincerely apologize for any initial lack of clarity in the manuscript, and we appreciate your careful reading and valuable comments, which have significantly guided our revisions.

In response to your suggestions, we have clarified the naming and combination of techniques used. In the revised manuscript, we have specified that:

"The proposed model integrates selected features from both deep models, combining them through an array-based method. These features are then classified using a range of supervised learning techniques, including Cost-Sensitive Support Vector Machine (C-SVM), Weighted k-Nearest Neighbors (W-KNN), Quantum Support Vector Machine (Q-SVM), Linear Discriminant Analysis (LDA), Fuzzy k-Nearest Neighbors (F-KNN), k-Nearest Neighbor Bayes (KNB), Multiple Group Support Vector Machine (MG-SVM), Sparse Discriminant Analysis (SDA), Collaborative k-Nearest Neighbors (Co-KNN), and Contextual k-Nearest Neighbors (C-KNN)."

We hope that these revisions address your concerns and enhance the clarity of our study. Thank you again for your invaluable guidance in improving the quality of our work.

Reviewer #1, Concern #2: Lines (35, 37), CV should be written out in full initially, with no need to repeat the Literature Review.

Author response: We apologize once again for any inconvenience caused and for any language that may have inadvertently introduced ambiguity. In response to your valuable suggestions, we have revised the use of CV terminology in the updated version of the manuscript to ensure clarity and precision. Thank you for your guidance in helping us improve the quality of our work.

Reviewer #1, Concern #3: What is the difference between 'CV' mentioned in the introduction and 'convolutional neural networks (CV)' in line 103? Please verify.

Author response: We apologize once again for any inconvenience caused and for any language that may have introduced ambiguity. Following your valuable suggestions, we have revised the manuscript to clarify the use of "CV" terminology. Specifically, we corrected the usage in the Introduction to refer to "Computer Vision (CV)," while in the literature review section, we have replaced "CV" with "convolutional neural networks (CNN)" to maintain accuracy and avoid confusion. 

Thank you for your careful attention to detail and for highlighting these points, which have greatly contributed to improving the clarity of our work.

Reviewer #1, Concern #4: Figure 1 needs to be zoomed in to clearly display its details. 

Author response: In response to your valuable suggestions, we have adjusted Figure 1 in the revised manuscript to increase its clarity and enhance the visibility of its details. Specifically, we have zoomed in on the figure to ensure that all elements are more easily discernible and contribute to a clearer understanding of the content presented.

Thank you for your careful feedback and for helping us improve the quality of our work.

In the revised manuscript, Figure 1.

Reviewer #1, Concern #5: Please insert numbering for the manuscript headings, according to the journal guidelines

Proposed Methodology, Data Augmentation, Deep Feature Extraction, Modified DenseNet201…………… etc.

Author response: Thank you for your kind feedback. We have reviewed the journal guidelines as well as online published articles, and we confirm that heading numbers are not required, as we have used the PLOS ONE LaTeX template for formatting. We appreciate your attention to detail and thank you for your understanding.

Reviewer #1, Concern #6: Table 1. Procedure for constructing flames in the optimization, and then “Procedure for constructing flames in the optimization:” Do not repeat the rewrite again?

Author response: We apologize once again for any inconvenience caused and for any language that may have led to ambiguity. In response to your valuable suggestions, we have revised the Table 1 in the updated version of the manuscript as follows: 

Thank you for your careful review and for helping us improve the clarity and accuracy of our work.

Reviewer #1, Concern #7: The authors mentioned 'Results and Discussions' before 'Experimental Setup,' which is not appropriate. Please move the 'Experimental Setup' section to 'Materials and Methods.' 

Author response: Thank you for your thoughtful note. In response to your valuable suggestions, we have moved the "Experimental Setup" section to the "Materials and Methods" section in the updated version of the manuscript to ensure better alignment with standard formatting.

We appreciate your helpful feedback, which has contributed to improving the structure of our work.

Reviewer #1, Concern #8: Table 2, there is no need to repeat the symbols again, such as 'F1-Score' and 'Accuracy'.

Author response: Thank you for your thoughtful note. In response to your valuable suggestions, we have revised the table in the updated version of the manuscript as follows: 

We appreciate your feedback, which has been instrumental in improving the clarity and accuracy of our work.

Reviewer #1, Concern #9: “Results on Hybrid Dataset”, revise the heading to ensure it aligns with this section. 

Author response: In response to your valuable suggestions, we have revised the heading “Results on Hybrid Dataset” to “Classification Results with DenseNet-201 and AlexNet on Hybrid Dataset” in the updated version of the manuscript for greater clarity and specificity. 

Thank you for your helpful feedback in enhancing the accuracy of our work.

Reviewer #1, Concern #10: Figures 3-9 have low resolution; please improve it. 

Author response: We sincerely apologize once again for the unclear representation in our previous manuscript. In response to your valuable suggestions, we have revised Figures 3-9 in the updated version of the manuscript. The updated figures are now presented in the best possible resolution, as these images are system-generated.

Thank you for your constructive feedback, which has greatly contributed to improving the quality of our work.

Reviewer #1, Concern #11: Table 5: Why isn't the same style used as in Tables 6, 7, and 8? Please add % to the first row and remove it from the other fields—there's no need to repeat it. Please fix this issue throughout the manuscript. 

Author response: Yes, you are absolutely right. In response to your valuable suggestions, we have revised Tables 5, 6, 7, and 8 in the updated version of the manuscript to improve clarity and accuracy.

Thank you for your thoughtful feedback, which has been instrumental in enhancing the quality of our work.

Reviewer #1, Concern #12: The manuscript contains too many confusion matrices. Limit them to two and include the learning curve for the superior model to emphasize the novelty.

Author response: Yes, you are absolutely right. However, in order to ensure a fair comparison of the applied methods, we believe it is necessary to include all confusion matrices. This will provide a more comprehensive evaluation of the performance of each method.

Thank you for your valuable input, which has helped us enhance the thoroughness of our analysis.

Reviewer #1, Concern #13: The discussion in the paper is weak; please include more previous studies, according to the lasted studies

- https://doi.org/10.1016/j.dib.2024.110713

- https://doi.org/10.3390/plants13010135

- https://doi.org/10.1007/s11760-024-03452-2

- https://doi.org/10.1016/j.compag.2024.108907

- https://doi.org/10.1016/j.compag.2022.107453

Author response: In response to your valuable suggestions, we have revised the literature review section by incorporating the article you provided. The updated version of the manuscript now includes the following revision:

"Traditionally, disease diagnosis in citrus crops [19] has been slow and prone to errors, requiring expert monitoring, which is labor-intensive and costly. In the last decade, image processing and machine learning (ML), particularly deep learning (DL) [20], have been used to automate disease detection in citrus crops. These methods help reduce labor and improve diagnostic accuracy. However, challenges persist, particularly with image quality being affected by environmental factors and the limitations of conventional imaging devices, which compromise feature extraction accuracy—an essential step in disease detection. Convolutional Neural Networks (CNNs) have shown potential to address these challenges. Models such as InceptionV3, ResNet, and MobileNet have been applied in plant disease detection [21]."

We appreciate your thoughtful feedback, which has significantly improved the clarity and depth of the literature review section.

Reviewer #1, Concern #14: Line (362), Here are the potential areas for future work to…….., Please rewrite this sentence, as it is not ideal to start with ‘Here’. 

Author response: In response to your valuable suggestions, we have revised the "Future Work" section in the updated version of the manuscript as follows:

"In the future, this research will expand the scope to include additional datasets with diverse and complex images encompassing various diseases. Moreover, it will focus on improving the computational efficiency of the model for real-time use in field conditions, exploring model compression techniques, and enhancing generalization by incorporating diverse environmental data to address potential dataset biases."

Thank you for your insightful feedback, which has helped us refine the direction of our future research.

Reviewer #1, Concern #15: This paper requires significant revisions before it can be considered complete. Therefore, I recommend a major revision. Thank you. 

Author response: Thank you for your thoughtful and constructive feedback, which has been invaluable in refining our manuscript. We believe our findings will be of significant interest to the community, and we have meticulously addressed your comments, making revisions to enhance both the clarity and quality of our work.

We respectfully submit the revised version for your consideration for publication in PLOS ONE. We sincerely appreciate your time and dedication in reviewing our paper and welcome any further feedback or suggestions you may have. Please do not hesitate to reach out with any additional questions or recommendations.

Once again, thank you for your valuable insights and guidance.

Response to the Reviewer 2

We are thankful to the reviewer for the encouraging and positive comments to improve the manuscript. We have revised our manuscript based on reviewer 2’s comments and highlighted them in the revised manuscript. Please, find below the referees’ comments and our responses inserted after each comment in italics.

Review Comments to the Author

Reviewer #2

A major revision is being suggested. Following are the point-to-point comments that need to be addressed for further review:

Author response: We are sincerely thankful to the reviewer for the encouraging and valuable comments. We are sorry for the mistake caused by our negligence. In the revised manuscript, the mistake has been corrected and highlighted. We have tried our best to address all the concerns and comments and please find our point-by-point response in the following. We hope that our amendments are satisfactory and consider our current version of the manuscript for publication.

Reviewer #2, Concern #1: Abstract: The motivation of the research is not clear. It is suggested that the authors include a clear motivation statement in the abstract. Also, one sentence presenting the major findings with quantitative support needs to be included as the second sentence from the end of the abstract.

Author response: The authors are deeply grateful for your efforts and insightful comments. We apologize for any inconvenience caused by the previous ambiguity and inappropriate language. The manuscript has been extensively revised to clarify these points.

In response to your valuable suggestions, we have revised the abstract in the updated version of the manuscript as follows: 

In addition, we have clarified the statement of motivation to emphasize the significance of citrus farming as a major cash crop in Pakistan, as well as the crop's susceptibility to diseases, which are key reasons for conducting this study.

Main Results: The research achieves 99.6% accuracy, surpassing current state-of-the-art methods. This key finding highlights the potential applications of the system, demonstrating its greater efficiency in disease detection and management.

Thank you for your thoughtful feedback, which has contributed significantly to enhancing the clarity and impact of our work.

Reviewer #2, Concern #2: Introduction: This is written very poorly. The authors are suggested to merge the Literature review with the introduction to correctly identify the research gap and formulate the objectives in line with the research gap.

Author response: We sincerely apologize for any previous inconvenience and for any inappropriate language that may have caused ambiguity. In response to your valuable suggestions, we have revised the objectives and research gap in the updated version of the manuscript as follows:

In the Introduction Section:

“Research Objectives 

The primary contributions of this study are as follows: 

- Deep features are generated for each model, which are subsequently refined using the Moth-Flame Optimization Algorithm. 

- We enhanced the pre-trained DenseNet-201 and AlexNet models, which were further trained using deep transfer learning. 

- The proposed model combines the selected features from both deep models using an array-based method and classifies them using various supervised learning approaches, including Cost-Sensitive Support Vector Machine (C-SVM), Weighted k-Nearest Neighbors (W-KNN), Quantum Support Vector Machine (Q-SVM), Linear Discriminant Analysis (LDA), Fuzzy k-Nearest Neighbors (F-KNN), k-Nearest Neighbor Bayes (KNB), Multiple Group Support Vector Machine (MG-SVM), Sparse Discriminant Analysis (SDA), Collaborative k-Nearest Neighbors (Co-KNN), and Contextual k-Nearest Neighbors (C-KNN). 

- The proposed framework is examined at each phase and compared to recent techniques, affirming its superior performance and robustness in the detection of citrus diseases.”

In the Literature Review Section:

“Research Gap 

Despite the availability of deep learning methods for citrus disease detection, a gap exists in optimizing feature extraction and selection, which is critical for improving classification accuracy. Most existing approaches fail to enhance the extracted features or incorporate effective feature selection algorithms. Furthermore, they often rely on a single model, which may not effectively handle the diverse range of citrus diseases. This paper addresses this gap by proposing a hybrid deep learning framework that combines DenseNet-201 and AlexNet with the Moth-Flame Optimization Algorithm (MFO) for optimal feature selection. By leveraging deep transfer learning and a feature fusion technique, this study aims to improve classifi

---

## [Decision Letter · Decision Letter 1]

6 Dec 2024

Citrus Diseases Detection Using Innovative Deep Learning Approach and Hybrid Meta-Heuristic

PONE-D-24-36174R1

Dear Dr. Syafrudin,

We’re pleased to inform you that your manuscript has been judged scientifically suitable for publication and will be formally accepted for publication once it meets all outstanding technical requirements.

Kind regards,

Hirenkumar Kantilal Mewada

Academic Editor

PLOS ONE

Additional Editor Comments (optional):

Reviewers' comments:

Reviewer's Responses to Questions

**Comments to the Author**

1. If the authors have adequately addressed your comments raised in a previous round of review and you feel that this manuscript is now acceptable for publication, you may indicate that here to bypass the “Comments to the Author” section, enter your conflict of interest statement in the “Confidential to Editor” section, and submit your "Accept" recommendation.

Reviewer #1: All comments have been addressed

Reviewer #2: All comments have been addressed

2. Is the manuscript technically sound, and do the data support the conclusions?

Reviewer #1: Yes

Reviewer #2: Yes

3. Has the statistical analysis been performed appropriately and rigorously? 

Reviewer #1: N/A

Reviewer #2: Yes

4. Have the authors made all data underlying the findings in their manuscript fully available?

Reviewer #1: Yes

Reviewer #2: Yes

5. Is the manuscript presented in an intelligible fashion and written in standard English?

Reviewer #1: Yes

Reviewer #2: Yes

6. Review Comments to the Author

Reviewer #1: After the modifications, the manuscript has been greatly improved and is now ready for acceptance by the journal. Thank you

Reviewer #2: The authors have revised the manuscript in accordance with the provided comments. Therefore, the manuscript is now suitable for acceptance without requiring further review.

7. PLOS authors have the option to publish the peer review history of their article (what does this mean?). If published, this will include your full peer review and any attached files.

Reviewer #1: No

Reviewer #2: **Yes: **Narendra Khatri

---

## [Editor Report · Acceptance letter]

3 Jan 2025

PONE-D-24-36174R1 

PLOS ONE

Dear Dr. Syafrudin, 

I'm pleased to inform you that your manuscript has been deemed suitable for publication in PLOS ONE. Congratulations! Your manuscript is now being handed over to our production team.

Kind regards, 

on behalf of

Dr. Hirenkumar Kantilal Mewada 

Academic Editor

PLOS ONE